# Functional Fine-Tuning of Metabolic Pathways by the Endocannabinoid System—Implications for Health and Disease

**DOI:** 10.3390/ijms22073661

**Published:** 2021-04-01

**Authors:** Estefanía Moreno, Milena Cavic, Enric I. Canela

**Affiliations:** 1Department of Biochemistry and Molecular Biomedicine, Faculty of Biology, University of Barcelona, and Institute of Biomedicine of the University of Barcelona (IBUB), 08028 Barcelona, Spain; 2Department of Experimental Oncology, Institute for Oncology and Radiology of Serbia, Pasterova 14, 11000 Belgrade, Serbia; milena.cavic@ncrc.ac.rs; 3Centro de Investigación Biomédica en Red sobre Enfermedades Neurodegenerativas (CIBERNED), 28031 Madrid, Spain

**Keywords:** endocannabinoid system, cancer, cannabinoid receptor, homeostasis, metabolism regulation

## Abstract

The endocannabinoid system (ECS) employs a huge network of molecules (receptors, ligands, and enzymatic machinery molecules) whose interactions with other cellular networks have still not been fully elucidated. Endogenous cannabinoids are molecules with the primary function of control of multiple metabolic pathways. Maintenance of tissue and cellular homeostasis by functional fine-tuning of essential metabolic pathways is one of the key characteristics of the ECS. It is implicated in a variety of physiological and pathological states and an attractive pharmacological target yet to reach its full potential. This review will focus on the involvement of ECS in glucose and lipid metabolism, food intake regulation, immune homeostasis, respiratory health, inflammation, cancer and other physiological and pathological states will be substantiated using freely available data from open-access databases, experimental data and literature review. Future directions should envision capturing its diversity and exploiting pharmacological options beyond the classical ECS suspects (exogenous cannabinoids and cannabinoid receptor monomers) as signaling through cannabinoid receptor heteromers offers new possibilities for different biochemical outcomes in the cell.

## 1. Introduction

Although the biological components of the endocannabinoid system (ECS) are well known and have been explored in detail over many decades, its significance seems to enlarge with every new experimental study. The ECS employs a huge network of molecules (receptors, ligands, and enzymatic machinery molecules) whose interactions with other cellular networks have still not been fully elucidated. It has become evident that its historical role in pain alleviation is just the tip of an enormous iceberg of translationally significant information that can be derived from the so-called endocannabinoidome. The ECS is involved in the modulation of a large amount of cognitive and physiological processes involved in the homeostatic regulation of the body. The role and mechanism by which the ECS is involved in the regulation of metabolism is not fully known, but its action is in large part through cyclic AMP/receptor activation-related pathways activated by cannabinoid ligands [1,2]. Endogenous cannabinoids are molecules with the primary function of control of multiple metabolic pathways. They are pre-synthesized and stored in cellular vesicles and released upon endogenous and exogenous stimuli to regulate internal homeostasis. Their targets include classical cannabinoid receptors that belong to the G-protein coupled receptor (GPCR) family as well as their various heteromers (see below), contributing to the complexity of the ECS [3]. Cannabinoid ligands also act through various non-canonical pathways [4], employing secondary messenger systems (changes in intracellular Ca^2+^ levels, activation of protein kinases) thus preferentially triggering alternative outcomes depending on the initial stimuli. This work aims to contribute to the growing burden of evidence that the ECS might be significantly more used as a pharmacological target for various metabolic disorders, despite carrying a historical label of being legally and ethically compromised.

## 2. Expression of Cannabinoids and Cannabinoid Receptors in Human Cells and Tissues

Endocannabinoids and endocannabinoid-like molecules (ECLs) are expressed by many cells in the human organism and their levels reflect the metabolic changes necessary for the homeostatic balance as well as response to pathological stimuli. There is high diversity within these molecule groups, both structurally and in the type of receptor they can stimulate, as well as the non-canonical biochemical pathways they affect [5]. Conventional endocannabinoids include anandamide (AEA) and 2-arachidonoylglycerol (2-AG), the ECLs *N*-acylethanolamines (NAEs) like *N*-palmitoyl-, *N*-oleoyl and *N*-linoleoyl-ethanolamine (PEA, OEA and LEA), and 2-acyl-glycerols (2-AcGs) like 2-oleoyl and 2-linoleoyl-glycerol (2-OG and 2-LG), prostaglandin ethanolamides, prostaglandin glycerol esters and omega-3 endocannabinoids that are derived from docosahexaenoic acid (DHA) and eicosapentaenoic acid (EPA) (like docosahexaenoyl ethanolamide (DHA-EA), docosahexanoyl-glycerol (DHG), eicosapentaenoyl ethanolamide (EPA-EA) and eicosapentanoyl-glycerol (EPG)) [6]. Along with all the corresponding metabolic enzymes and molecular targets they constitute a large network called the endocannabinoidome, or the expanded ECS [7]. The metabolic complexity of such a huge network of molecules needs to be carefully evaluated in each setting, which might be facilitated with the increasing power of “omics” methods and modern sequencing technologies.

There is a number of human receptors currently described in the literature that respond to cannabinoid ligands [3], but the most studied are the main cannabinoid receptors 1 (CB1R), coded by CNR1 gene and 2 (CB2R), coded by CNR2 gene [8,9] that belong to the GPCR family. Other receptors that respond to various cannabinoid ligands are G protein-coupled receptors 18 (*N*-arachidonyl glycine receptor, GPR18), GPR55, GPR119 and the transient receptor potential cation channel subfamily V members 1 and 2 (TRPV1 and TRPV2) [7], but this review will mostly focus on the analysis of classical cannabinoid receptors CB1R and CB2R. The mRNA expression in humans, specificity and significance of CB1R and CB2R was explored using the freely available interactive database the Human Protein Atlas (HPA) [10,11]. A schematic pictorial model of CB1R and CB2R expression in human organs and tissues according to the Tissue Atlas of the Human Protein Atlas database is presented on Figure 1.

Analysing human tissue specificity by combination of expression profile data from various sources on the mRNA and protein level using the HPA Tissue Atlas subproject, it was detected that CB1R is tissue enhanced in the brain (mRNA level), and in adipocytes, pituitary gland and the central nervous system (CNS) (protein level) (Figure 2a), while CB2R is tissue enriched in blood and lymphoid tissue (mRNA level) and present in variable protein levels in most tissues (Figure 2b).

Single cell-type specificity analysis using scRNA-sequencing data from human normal tissues and a large panel of cell lines and protein localization data derived from antibody-based profiling in the HPA Cell Type Atlas and the Cell Atlas showed that CB1R is cell-type enriched in Sertoli cells, mainly localized in the plasma membrane in different isoforms and additionally in the actin filaments. On the other hand, CB2R is found to be cell-type enhanced in B-cells, T-cells and alveolar cells type 2 and localized in the plasma membrane. The data from the HPA Blood Atlas that present transcriptomics analysis of human blood and cultured cell lines showed that CNR1 is group enriched in memory B-cells and naive B-cells, while CNR2 is group enriched in basophils, eosinophils, naive B-cells, memory B-cells and NK-cells. The significance of these receptors in cancer has been previously explored using the HPA Pathology Atlas [12] which employs information about the protein expression and correlation between mRNA expression and patient survival for 17 different types of human cancer. Although these data showed that these two receptors were generally not prognostically significant, CB1R has been found to be enriched in glioma and CB2R in testicular cancer. The existence of other receptors that respond to ECS ligands and the fact that GPCR receptors also have the ability to heteromerize giving rise to new receptor entities [13,14,15] multiplies the regulatory potential and diversity of the molecules involved in the transmission of ECS-related signals. Although specific cellular and tissue distribution of various heteromers in humans is still under intense investigation [13,16,17], their distinct pharmacological properties have already been detected.

## 3. Involvement of the ECS in Specific Physiological and Pathological Processes

Considering the diversity of ligands, the number of receptors and heteromers, their tissue and cellular localization, as well as the vast network of other molecules belonging to the ECS, it is not surprising that this system is involved in the regulation of numerous essential physiological and pathological pathways [4] (Figure 3).

The ECS ligands have primarily been used in pain alleviation, which is of great value as pharmacological studies have already been performed enabling their faster repurposing as has been previously suggested in other settings [16,17]. This approach has become especially important in the COVID-19 pandemic era when extreme global negative real-world effects were detected in many aspects of patient care management [18,19] and fast responses from the scientific community are needed. Psychotropic side effects of some ECS-related ligands should not be regarded as an obstacle, as it has been previously shown that exogenous ligands with these properties comprise only a small part of the milieu of potential ECS modifiers [20,21]. Currently explored directions include the investigation of drugs that do not pass the blood–brain barrier, using lower doses to limit potential side effects and allosteric modulators among others.

The metabolism of ECS depends in a great deal on redundant enzymatic cascades employed by other biologically active mediators which introduces a challenge in the interpretation of experimental data relating to the regulation of metabolism and homeostasis as well as their further application [1,2]. An overview of most recent literature data on the involvement of the ECS in various physiological and pathological processes is presented in Table 1.

### 3.1. Glucose and Lipid Metabolism

Several studies have shown a positive association of plasma endocannabinoids with markers of metabolic disorder and obesity [30,86,87,88,89,90]. These new findings show that the ECS acts as a regulator of metabolic homeostasis. The ECS, located at a central and peripheral level, regulates the commands issued by different brain regions, it regulates the communication between the brain and the periphery and adjusts the activity of every organ involved in lipid and glucose metabolism. Its overall function promotes energy intake and storage, however, when highly-caloric and palatable food is available, the anabolic consequences of ECS overactivation may promote obesity and metabolic disorders, such as hypertension, hypertriglyceridemia and insulin resistance leading to the development of metabolic syndrome and type 2 diabetes [30,91]. The ECS participates in different tissues in the regulation of lipid and carbohydrate metabolism to regulate metabolic homeostasis and their respective metabolic disorders.

A complete ECS has been found in both murine and human adipocytes [32,88,92]. The CB1R is highly expressed throughout the CNS in neurons regulating feeding, energy expenditure, and reward, as well as in peripheral organs that are critical for metabolic homeostasis [92,93,94]. There is evidence that the CB2R is also expressed neuronally, and some authors have found CB2R to be expressed in differentiated adipocytes, while others failed to find significant expression [26,27,95,96]. In 2003, two unrelated studies unravelled the presence of functional CB1Rs in white adipocytes [24,32,97]. This discovery led the way to explore the presence and function of this receptor in peripheral non-neuronal tissues (adipose tissue, pancreas, liver, gastrointestinal tract, and skeletal muscles).

In the adipose tissue, the CB1R stimulation increases the activity of the lipoprotein lipase (LPL), promoting the hydrolysis of triglycerides and their subsequent uptake [22]. Additionally, CB1R stimulation enhances fat storage within adipocytes through activation of lipogenic enzymes and inhibition of the activity of the 5′-AMP-activated protein kinase (AMPK) [23]. Aside from favouring lipogenesis, CB1R regulates also adipogenesis by increasing the expression of the nuclear receptor peroxisome proliferator-activated receptor-gamma (PPARγ), promoting adipocyte differentiation [98]. Arachidonoylethanolamide (anandamide, AEA) can also act as a PPARγ agonist, amplifying the adipogenesis caused by the ECS [26,27]. Conversely, the pharmacological inhibition of CB1R induces fatty acid oxidation, mitochondrial biogenesis via increased expression of the endothelial nitric oxide synthase [99] and the differentiation of white adipocytes into beige adipocytes [100]. CB1R regulates white adipose tissue (WAT) expansion, maintenance of white adipocyte phenotype and the development of insulin resistance and obesity. Thus, the results seen in vitro might be due to a direct peripheral action of endocannabinoids, although the sympathetic nervous system (SNS) is also involved in these responses [30]. In addition to cannabinoid receptors (CBRs), fat cells and adipose tissue express the enzymatic system to produce and degrade locally the endogenous cannabinoids [26,27,89,101]. Currently, the non-psychotropic component of Cannabis Sativa, cannabidiol (CBD), affects both glucose and lipid metabolism through the action on various receptors as well as several metabolites in adipose tissue, pancreas, liver, and cardiac muscle [28]. CBD is able to block CB1R, producing anti-obesity effects and might be effective in relieving the symptoms of insulin resistance, type 2 diabetes and metabolic syndrome [28].

Activation of CB1R also stimulates glucose entry into fat cells. In human adipose cells, CB1R stimulation promotes glucose uptake, and this effect is mediated by translocation of the insulin-regulated glucose transporter type 4 (GLUT4) to the plasma membrane from the intracellular compartment. In addition, cannabinoid-stimulated glucose uptake in fat cells is mediated by the same molecular machinery that is responsible for insulin-induced glucose uptake, i.e., activation of PI3-kinase. In fact, inhibition of this enzyme completely disrupts the effect of CB1R activation on glucose uptake [24,25]. Other studies have proved that natural extract containing Δ^9^-Tetrahydrocannabinol (Δ^9^-THC) decreased the triacylglycerols (TAGs) content and improved the glucose uptake in the insulin-resistant in a concentration-dependent manner [29], by an enhanced GLUT4 and insulin receptor substrate 1 and 2 (IRS-2) gene expressions [28].

In the pancreas, endocannabinoids play an important role in the regulation of cell proliferation and classification of α/β cell during pancreatic islets formation, with an impact on programming of pancreatic glucagon and insulin secretion [30,31]. In the liver, activation of hepatic CB1R by endocannabinoids induces the expression of acetyl coenzyme-A carboxylase-1 (ACC1), fatty acid synthase (FAS) and sterol regulatory element-binding transcription factor 1 (SREBPF1), resulting in fatty acid synthesis and leading to hepatic steatosis [32], however, mice with hepatic CB1R deleted are protected from metabolic disorders such as dyslipidemia, hyperglycemia, insulin resistance and hepatic steatosis [32,34,102]. So, hepatic CB1R exerts an important role in the regulation of glucose and lipid metabolism. In the skeletal muscle, some studies suggest that the ECS can affect glucose homeostasis where CB1R activation decreases the glucose uptake, an effect that can be blocked by pharmacological inhibition of CB1R [30,95].

Chronic administration of selective CB1R antagonist Rimonabant in humans was successful at reducing body weight, fat mass and metabolic impairments related to obesity, such as diabetes and dyslipidemia [96]. In 2006, the compound was approved by the *European Medicines Agency* (EMA) as an anti-obesity therapy, but in 2008, its use was suspended, based on the fact that its benefits no longer outweighed its risks, taking into account that patients with an elevated risk of developing psychiatric disorders could not be identified. The fall of rimonabant obstructed future drug development aiming at the ECS and occasioned a profound controversy about the relevance of modulating the ECS in obesity and metabolic disorders. Since the major side effects of drugs like rimonabant were CNS related, one opportunity to move forward could be provided by CB1R antagonists that are unable to cross the blood–brain barrier [97]. Some of these drugs, such as the peripherally restricted CB1R inverse agonist JD5037 and the CB1R antagonist AM6545 have been shown to reduce obesity, reverse leptin resistance and improve dyslipidemia, hepatic steatosis and insulin resistance in genetically and diet-induced obese mice [30,33,34,35]. Subsequent studies have shown that JD5037, is even more effective in improving metabolic parameters in rodent models of obesity/diabetes and has hypophagic effects by reversing leptin resistance [33], abolishes obesity-induced hepatic insulin resistance [101] and preserves beta-cell function [103]. These results increase the prospects that CB1R blockade may still be a viable option to combat dysmetabolism and may move to clinical testing in the future [104]. The clinical studies with CB1 agonists, partial agonists, inverse agonists and neutral antagonists clearly point out the CB1R as a potential effective target for the treatment of obesity [2,102,105,106,107]. Another way of modulating CB1R activity is represented by compounds that could be developed by studying recently identified endogenous allosteric inhibitors of CB1R. Hemopressin, pepcans and the neurosteroid pregnenolone have been identified presenting that function [108,109,110]. Hemopressin reduces food intake without causing any obvious adverse side effects [2,111,112]. Nevertheless, further studies are needed in order to confirm that these effects are due to the direct action of hemopressin on CB1R [113], whereas pregnenolone binding to CB1R does not modify the binding of agonists, but reduces body weight gain in diet-induced obese mice and it does not induce anxiety [30,109].

### 3.2. Food Intake Regulation

The past few years have seen a significant increase in the number of human studies trying to understand the role of the ECS in the regulation of eating behaviour and metabolism. Endocannabinoids can be detected in the bloodstream and their assessment from blood samples is a simple strategy used for the study of the ECS. The ECS participates in the development of preference for the consumption of certain foods, even in humans [114], it modulates olfactory responses and taste [115,116] and controls metabolic changes associated with food intake. Therefore, the type of diet consumed affects endocannabinoid levels and ECS action [117]. The presence of fat in the oral cavity induces the production of jejunal endocannabinoids, which will increase fat intake, and gastric CB1R activation leads to ghrelin secretion, which increases fat-taste perception and promotes fat intake [30,36].

Various studies have therefore attempted to establish a functional connection between circulating endocannabinoids and feeding behaviour. Both normal-weight and obese subjects have a peak in plasma AEA before a meal, but not 2-arachidonoylglycerol (2-AG), implying that AEA may act as a meal initiator signal in humans [37]. Nevertheless, when the impulse for food is related to its palatability as opposed to hunger, others authors have observed an increase in plasma 2-AG in both healthy and obese subjects [38,39]. This implies that AEA and 2-AG may have different roles in regulating the eating behaviour, the former acting to initiate the intake of calories, and the latter to maintain the intake beyond satiety [30]. Close relationships were also found between the ECS and hormones affecting energy balance regulation, like glucocorticoids, ghrelin and leptin [118]. Endocannabinoid levels in the WAT are negatively regulated by insulin [23] and leptin [119]. This effect might be lost under insulin or leptin resistance, thus promoting ECS overactivity and fat accumulation. Treatment of diet-induced obese mice with JD5037 diminishes leptin production and secretion by adipocytes. The consequently diminished leptinemia reverses leptin resistance, resulting in a decrease in body weight and food intake [33]. Additionally, AM6545 is shown to reduce obesity, reverse leptin resistance and improve dyslipidemia, insulin resistance and hepatic steatosis in obese mice [30,33,34,35].

There are studies that show a relationship between circadian regulation and ECS signaling in the CNS. It appears that the circadian clock of the CNS regulates ECS via modulating synthesis, degradation and transport mechanisms so that ECS has a possible modulatory role [120]. Levels of endocannabinoids in plasma and cerebrospinal fluid may vary depending on the race [121] and endocannabinoids also change across the 24 h sleep-wake cycle and sleep deprivation altering their levels, which is accompanied by increased hunger [122]. Sleep disturbances are known to be a risk of obesity [123], consequently, plasma AEA is increased in patients suffering from sleep apnea [30,124]. Recent studies have also increased our knowledge of the function of the ECS in the CNS. Endocannabinoids regulate appetite and food intake via activation of CB1R. It has been established in recent years that depending on the brain region and the location of CB1Rs, the consequences of their activation can be altogether different. The control of food intake by the ECS depends on whether CB1Rs are located on GABAergic or glutamatergic terminals [30,40]. With low doses of Δ^9^-THC, the suppression of glutamatergic transmission induces a rise of appetite. However, when the doses are higher, GABAergic transmission is disturbed, resulting in hypophagia [30,40]. This could explain earlier reports in humans where biphasic effects of cannabis and/or Δ^9^-THC were observed on food consumption depending on the dose used. Additionally, the lack of CB1R in dorsal telencephalic glutamatergic neurons prevents the development of food addiction-like behaviour [125].

Recent investigations have also shown a powerful emerging link between the ECS and another major player in metabolism and the gut microbiome (ensemble of genes, proteins, and metabolites provided by intestinal microorganisms). There are several instances of how lifestyle modifications (westernized diets, lack or presence of certain nutritional factors, physical exercise, and the use of cannabis) can regulate the inclination to develop metabolic syndrome by modifying the crosstalk between the ECS and the gut microbiome [126]. In fact, the consumption of food enriched in n-3 Poly-Unsaturated Fatty Acid (PUFA) diminishes anandamide levels and improves the lipid profile in obese subjects. Thus, higher consumption of n-3 PUFA in the diet might represent an effective approach to help prevent and treat metabolic disorders [41,42].

### 3.3. Immunity and Inflammation

Numerous components of the ECS function as key regulators of the immune system and the immune response. The ECS plays an important role in the migration of hematopoietic stem and progenitor cells. Endocannabinoids can stimulate the migration of human hematopoietic stem cells in a CBR-dependent manner [43,127]. CB2R is mainly found in cells of the immune system and has an important role as a modulator of immune function [92,93,94]. CB2R participates in the retention of immature B cells in the bone marrow [44] producing a significant decrease in chemokine receptor type 4 (CXCR4) in bone marrow cells treated with the agonist CP55940 [43,45]. CB2R is involved in the inhibition of lymphocyte recovery following bone marrow transplantation (BMT) [47]. The ECS also participates in the regulation of mature immune cell trafficking and effector cell functions, and endocannabinoids play a fundamental regulatory role in the function of intestinal neutrophils. The transporter P-glycoprotein (P-gp) secretes endocannabinoids into the intestinal lumen counteracting the pro-inflammatory actions of the neutrophil chemoattractant eicosanoid hepoxilin A3. Furthermore, the anti-inflammatory actions of P-gp are mediated by CB2R on neutrophils [43,128], and CB2R deficiency intensifies acute neutrophils mobilization to sites of inflammation [129]. Both murine and human macrophages and microglial cells, express the CB1 and CB2 receptors [43,130,131,132,133,134,135], CB2R is involved in the inhibitory role of tumor-associated macrophages [136]. 2-AG, via activation of CB2R, may act as a chemotactic substance capable of recruiting dendritic cells and their precursors during the innate immune response [43,46]. The ECS also participates in the regulation of adaptive immunity. Although T-cells express fewer CB2R than other immune cells, it has been demonstrated that stimulation of T-cells can upregulate the expression of CB2R [45,50,137], and stimulate CB1R expression [49]. In vitro studies showed that cannabinoids inhibit T-cell activation acting through CB2R and other receptors [47,48,49]. Both CB1R and CB2R reduce interleukin 2 (IL-2) synthesis in T-cells [138]. ECS has a role in a mature B-cell function, indicating that anandamide, via CB2R, generates dose-related immunosuppression in plaque-forming cell assays of antibody formation [43,139]. All these instances reveal that the ECS is a key regulator of the immune system.

Indeed, it is widely documented that perturbations in endocannabinoids levels, along with modifications in all members of the ECS, take place in many chronic inflammatory-associated conditions, including cancer, diabetes mellitus, atherosclerosis, cardiovascular, chronic airway, inflammatory bowel, autoimmune and neurodegenerative diseases [52,54,66]. Prior studies highlighted the role of endocannabinoids as major suppressors of chronic inflammation [51]. In these studies, the immunosuppressive effects exerted by activation of the endocannabinoid signalling in leukocytes have been associated to the modulation of: production of inflammatory cytokines and other endogenous anti-inflammatory or pro-inflammatory mediators; chemotaxis and inflammatory cell recruitment; and immune cell proliferation, differentiation and apoptosis [50]. However, various studies have shown that depending on the context, these molecules can also perform pro-inflammatory actions. Especially, AEA appears to elicit mainly anti-inflammatory effects by decreasing T and B cell proliferation, while 2-AG exhibits proinflammatory and anti-inflammatory functions [50,51,52,64].

Clinical trials with the selective CB1R antagonist Rimonabant documented an augmented glucose uptake and adiponectin production causes a reduction of systemic levels of pro-inflammatory cytokines and enhanced glucose tolerance [52,53,54,55]. Instead, CB2R activation has been reported to enhance fat tissue inflammation and insulin resistance [54,58,59]. These recent studies show that the pharmacological blockade of CB1R prevents metabolic dysfunction and β-cell loss, while reducing body mass and mortality rate [52,140]. Thus far, abnormal endocannabinoid signalling, either due to their excessive production or to up-regulation of CB1R, which exerts damaging effects in diabetes, is considered a pathogenic factor in this inflammatory disease.

Given the critical role of the ECS as a main immunomodulatory player in the brain, it is thought that the neuroprotective effects of drugs based on endocannabinoids for neurodegenerative diseases are primarily mediated by decreasing the neuroinflammation, and thus normalization of key processes known to compromise neuronal homeostasis and survival, like excitotoxicity, oxidative stress and apoptosis. The majority of these immunoregulatory effects are attributable to the activation of CB2R, expressed both on microglia and brain-infiltrating immune cells, although there is also evidence for the involvement of CB1R in mitigating the immune response, as in the case of traumatic brain injury, Multiple Sclerosis (MS) and Alzheimer’s disease (AD) models [52,141,142,143]. Rising the levels of AEA and 2-AG in the brain with the use of inhibitors of the main degrading enzymes, fatty acid amide hydrolase (FAAH) and monoacylglycerol lipase (MAGL), has been documented as an effective strategy to control the immune response in different models of MS, Huntington’s disease (HD) and AD [52,58].

Endocannabinoids modulate nociception by reducing sensory neuron excitability and controlling the transmission of nociceptive signals to the CNS. CBD is effective in neuropathic and inflammatory pain in rodents [60,61], by modulating targets involved in the control of nociception, including inhibition of AEA enzymatic hydrolysis and indirect activation of CBRs; activation/desensitization of TRPV1 and TRPA1 channels [60,61]; activation of the 5-HT1AR; and inhibition of equilibrative nucleoside transporters (ENTs), which causes increases of adenosine signaling, analgesia, and inhibition of inflammation. Δ^9^-THC strongly reduces nociception in animal models of acute, visceral, inflammatory, and chronic pain [59,60]. It must be considered the interplay between the metabolism of endocannabinoids and prostanoid systems (both bioactive lipid systems). At the molecular level, the consequences of increased endocannabinoid levels in pain are not clear because this can result in analgesic or pro-analgesic effects. In fact, endocannabinoids can act beneficially through CBRs activation or otherwise have detrimental effects through arachidonic acid production and pro-analgesic prostaglandins (PGs), such as prostaglandin E2 (PGE_2_). Formation of prostaglandin glycerol esters (PG-G) and prostaglandin ethanolamides (PG-EA) by cyclooxygenase (COX)-2 metabolism of endocannabinoids could also lead to analgesic or pro-analgesic effects depending on the bioactive metabolite produced [144].

### 3.4. Respiratory Health and Diseases

Respiratory health depends a great deal on the airway epithelial cells that represent a physical and biochemical barrier to exogenous and endogenous pathogens. The integrity of ECS components in human airway epithelial cells has been reported as one of the key metabolic necessities for healthy lungs and physiological response to damaging stimuli [145]. The endocannabinoid AEA has the ability to increase the permeability of these cells by bioactive arachidonic metabolite formation inside the cells, rather than using canonical pathways [62]. 2-AG has also been characterized as an important source of lung prostaglandins which metabolizes into leukotriene B_4_ and C_4_ by neutrophils and eosinophils, providing a connection between the ECS and the prostaglandin system [63].

As CB1R and CB2R are in many cells of the immune system (eosinophiles, monocytes), their involvement in inflammatory and other immune-related events in the lungs is not surprising. CB2R agonists have been known to contribute to anti-inflammatory responses upon various stimuli leading to the inhibition of leukocyte recruitment and secretion of pro-inflammatory cytokines as TNF-*α*, interleukin 1*β* (IL-1*β*), IL-6, etc., and the reduction of the formation of reactive oxygen species [65]. The prolonged presence of lipopolysaccharides on the membranes bacteria elicits an increase in 2-AG production in mast cells due to activation of toll-like receptor 4, so CB2R activation can produce so-called endotoxin tolerance. [137] These processes are metabolically mediated by the activation of 5′ AMP-activated protein kinase, downregulation of anabolic processes and upregulation of oxidative phosphorylation [66]. An interplay between viral infections and CB2R has also been described in the lungs which enhances the ECS’s contribution to the physiological response to various respiratory infections [146]. On the other side, upregulation and activation of CB1R usually enhance oxidative stress and inflammation, although opposite effects have also been described [147]. In neutrophils, activation of CB1R might presynaptically inhibit the cholinergic transmission providing protection from airway inflammation and possible lung damage [67]. Inhibitors of MAGL and FAAH have also been described to induce a downregulation of TNF-*α*, PGE2, COX-2, inducible nitric oxide synthase (iNOS), and lead to a general anti-inflammatory metabolic state, due to a change in levels of fatty amino acids and endocannabinoids [66,67]. The roles of TRPV1 and GPR55 still remain to be elucidated in detail, although their activation has been linked to the release of proinflammatory cytokines, individually and in heteromer combinations [148].

The use of exogenous cannabinoid sources via inhalatory pathways has been linked with the onset of pathological respiratory metabolic symptoms such as those characteristic of asthma, pulmonary fibrosis and allergies [149]. The involvement of the ECS in these diseases is extremely complex, as there is a constant interchange of metabolites that target a variety of receptors present on many cells in the airways, balancing between a physiologically necessary response and pathological inflammation. In asthma and pulmonary fibrosis, as well as in nicotine-induced fibrosis, CBD has shown the potential to countermand inflammation via canonical pathways [150,151]. CB1R interacts with iNOS as well, and their simultaneous targeting using hybrid CB1R/iNOS inhibitors has shown potential for combating fibrosis [147]. The potential of ECS metabolic pathways has also been explored in the current SARS-CoV-2 pandemic, as an anti-inflammatory strategy during the cytokine storm phase, and in combination with anti-viral therapeutics [152]. CBD was found to effectively inhibit the JAK-STAT axis and block the production of type I interferons contributing to potentially lowering the severity of the respiratory disease, while being well-tolerated and safe [153]. Both exogenous and endogenous cannabinoids continue to be explored as plausible options for the SARS-CoV-2 crisis [154].

### 3.5. Cancer

The involvement of ECS components in the pathogenesis of tumors and anti-cancer treatment has been explored extensively using in vitro and in vivo models, as well as in clinical studies, many of which are still ongoing [12,155,156].

Reprogramming of vital metabolic pathways represents one of the key moments in the development and later progression of cancer [157]. New studies suggest that cell metabolism may be altered very early during tumorigenesis contributing as a driver event rather than appearing as a consequence [158]. Some of the paramount metabolic pathways implicated in carcinogenesis include glycolysis, glutaminolysis, metabolism of lipids and nucleotides, formation of reactive oxygen and nitrogen species by mitochondrial metabolism, inflammation, etc. [158,159,160]. As there is a complicated, multilateral exchange of metabolites between various pathways in the pre-cancerous cell, the role of the ECS in cancer development is ambiguous. Considering the environment that enables cellular transformation, many cells of the innate and adaptive immune system that express cannabinoid receptors CB2R and GPR55 and respond to endogenous cannabinoids invade the neoplastic cell with the aim to control its growth [76,161]. In this setting, MAGL has been detected as an important player, which is not surprising as it regulates the metabolism of long-chain fatty acids and contributes to the cancer-related reprogramming of lipid metabolism [136,162]. On the other hand, consumption of exogenous cannabinoid sources has been considered to have a pro-cancerous effect, mostly via inhalation of other carcinogens that interfere with the metabolism of nucleotides and lead to the production of reactive oxygen species [163].

The involvement of ECS once a cancerous state is already present needs to be carefully evaluated as the effects depend greatly on the type of cancer cell [12]. In hormone-sensitive BC, it has been shown that the endocannabinoid anandamide has the ability to block the cell cycle progression acting through CB1R [71,72,164], while the activation of CB2R inhibited chemotaxis due to the presence of CB2R-CXCR4 heteromers [165]. In aggressive high-grade tumors as HER2-positive metastatic BC, it has been shown that CB2R is overexpressed, and treatment with Δ^9^-THC, THC 14 and selective agonists for CB2R has shown great potential [155,156,166,167]. Various heteromers have also been explored in HER2+ BC (HER2-CB2R, CXCR4-CB2R, GPR55-CB2R) as anti-cancer targets and as prognostic biomarkers [13,14,84,168], and the endocannabinoid cannabidiol has been shown to affect BC cell proliferation and invasiveness [169]. In glioma, it has been shown that signaling through CB1R and CB2R, which are over-expressed in these tumor cells, with the involvement of de novo synthesis of ceramide induces apoptosis [73,74,155,170]. The endocannabinoids AEA and 2-AG, as well as cannabidiol and Δ^9^-THC, have been connected with the inhibition of in vitro proliferation, migration and invasiveness of glioma cells using canonical pathways [73,74,75,171]. In vivo and ex vivo studies have confirmed that cannabinoids as Δ^9^-THC, WIN-55,212-2 and JWH-133 have the ability to slow down tumour growth and prolong the survival in rats and tumor cells obtained from biopsies [171,172]. It is evident that the activation of canonical pathways by various cannabinoids leads to a functional fine-tuning of pathways essential for cellular proliferation, apoptosis, and angiogenesis. In gastrointestinal malignancies, high expression levels of CB2R have been linked with poor prognosis and aggressiveness through the AKT/GSK3β signaling axis, while low levels of CB1R are more present in high-grade and more invasive GI tumors [77,79]. MAGL which is essential for the metabolism of endogenous cannabinoids has also been linked with aggressiveness in colorectal cancer (CRC) [173]. Beneficial activation of apoptotic pathways in CRC via canonical (CB1R, CB2R, *PPARγ*) and non-canonical ECS-related pathways has been described for both endogenous, plant-derived (Δ^9^-THC, CBD) and synthetic cannabinoids (HU-331, CP 55,940) through the inhibition of RAS–MAPK and PI3K–AKT and increased ceramide synthesis among other [78]. In prostate cancer, over-expression of anandamide as well as CB1R, CB2R and GPR55 receptors has been linked with poor prognosis [81]. On the other hand, a variety of phyto-, endo- and synthetic cannabinoids, as well as MAGL inhibitors have shown potential in the modulation of pathways important for prostate cancer cell survival, migration and invasiveness [69]. Interference with the regulation of pathways employing adenylyl cyclase, protein kinase A, EGFR, as well as the presence of CB2R-CXCR4 in prostate cancer cells leads to the reduction of the cells’ invasive potential and offers strategies for combating metastatic disease [174,175]. CBRs are generally over-expressed in lung cancer inducing favourable effects on patient survival upon stimulation acting in part through ERK, PI3K, p38 MAPK, Akt, EGFR and ceramide-related pathways connected with apoptosis and epithelial-to-mesenchymal transition [84]. As lung cancer is an example of the success of modern molecular targeted approaches and immunotherapy, the interaction between ECS components with the metabolic processes induced by these therapies (DNA repair, epithelial-to-mesenchymal transition, immunomodulation) has also been suggested [83,164,176,177].

In many situations, tumours become or remain resistant to conventional therapies, or serious adverse events hamper such treatment, so many plant-derived alternatives have been explored as supplements or alternative medicines [178,179,180]. ECS components have shown great potential in this setting, as a boost or in combination with standard therapeutics and also in battling side effects as they interfere with the underlying driver metabolic mechanisms [168]. Besides the direct effects of the ECS on cancer cells and microenvironment, it has also been implicated in the proliferation and differentiation of embryonic and adult stem cells [181]. Most of these effects are mediated by the PI-3K/AKT and IL-1 signaling related to the TNF pathway, and have shown promising early results for the repression of cancer cell formation, invasiveness and metastasis [170]. As many anti-cancer drugs do not have the ability to eradicate cancer stem cells, using ECS co-targeting approaches might be useful for the prevention of resistance to therapy and cancer recurrence [182].

Although the ECS is explored in various anti-cancer scenarios, its clinical utility needs to be carefully evaluated in each specific tumor subtype as they differ in metabolic pathways that interact with ECS signals.

## 4. Conclusions

Maintenance of tissue and cellular homeostasis by functional fine-tuning of essential metabolic pathways is one of the key characteristics of the ECS. It is implicated in a variety of physiological and pathological states and an attractive pharmacological target yet to reach its full potential. Future directions should envision capturing its diversity and exploiting pharmacological options beyond the classical ECS suspects (exogenous cannabinoids and cannabinoid receptor monomers) as signaling through cannabinoid receptor heteromers offers new possibilities for different biochemical outcomes in the cell.

## Figures and Tables

**Figure 1 ijms-22-03661-f001:**
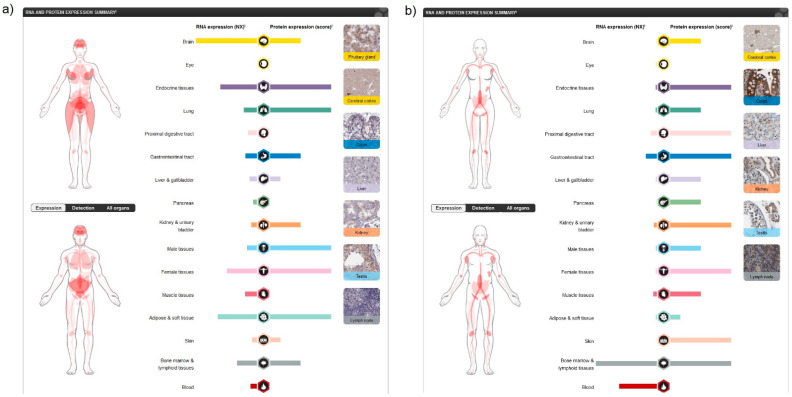
A schematic pictorial model of (**a**) CB1R and (**b**) CB2R expression in human organs and tissues according to the Tissue Atlas of the Human Protein Atlas database [10]. mRNA expression overview shows RNA as the consensus dataset based on a combination of three sources—RNA-sequencing data from internally generated Human Protein Atlas (HPA), RNA-sequencing data from the Genotype-Tissue Expression (GTEx) project and Cap Analysis of Gene Expression (CAGE) data from the Functional ANnoTation Of the Mammalian genome 5 (FANTOM5) project. When available protein data for which a knowledge-based annotation gave inconclusive results, no protein expression data were displayed. Nx—Normalized eXpression (resulting transcript expression values calculated for each gene in every sample).

**Figure 2 ijms-22-03661-f002:**
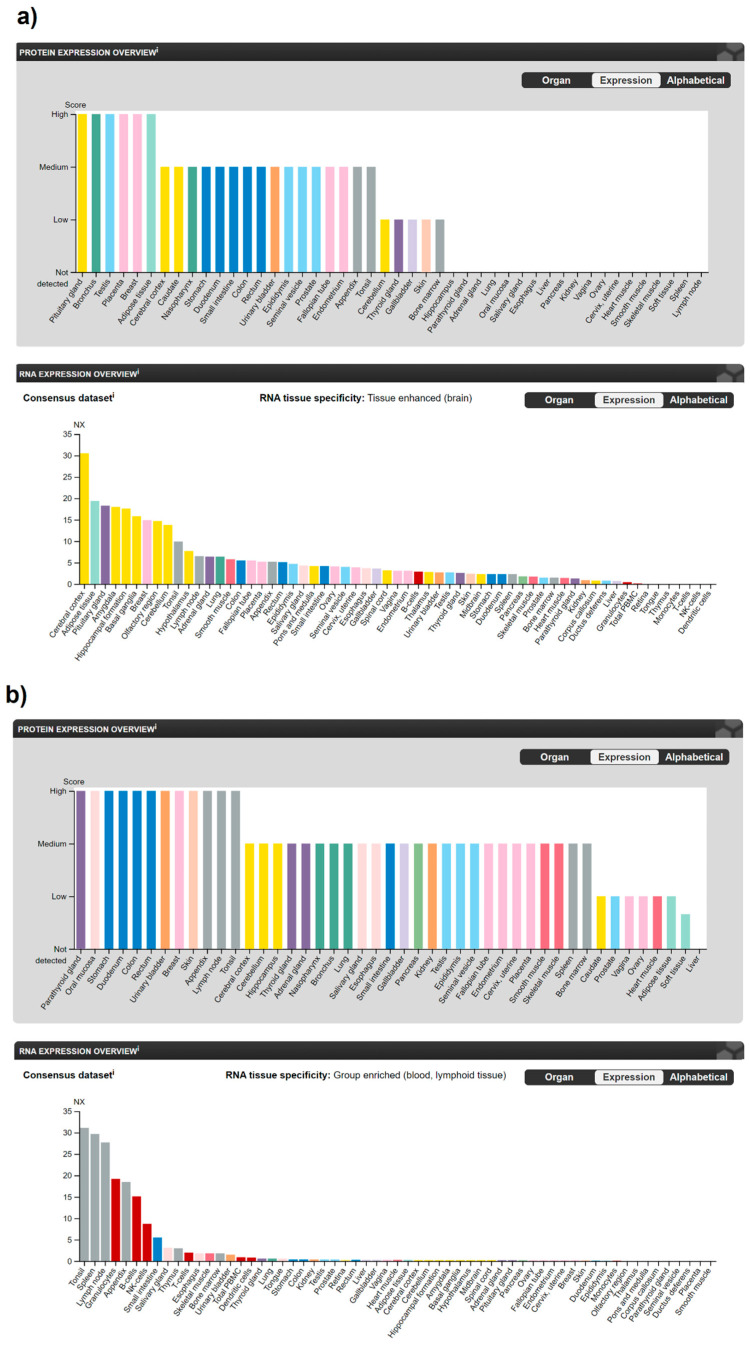
The expression of (**a**) CB1R and (**b**) CB2R in human tissues according to the Human Protein Atlas database [10]. mRNA expression overview shows RNA as the consensus dataset based on a combination of three sources—RNA-sequencing data from internally generated Human Protein Atlas (HPA), RNA-sequencing data from the Genotype-Tissue Expression (GTEx) project and Cap Analysis of Gene Expression (CAGE) data from the Functional ANnoTation Of the Mammalian genome 5 (FANTOM5) project. When available protein data for which a knowledge-based annotation gave inconclusive results, no protein expression data were displayed. Nx—Normalized eXpression (resulting transcript expression values calculated for each gene in every sample).

**Figure 3 ijms-22-03661-f003:**
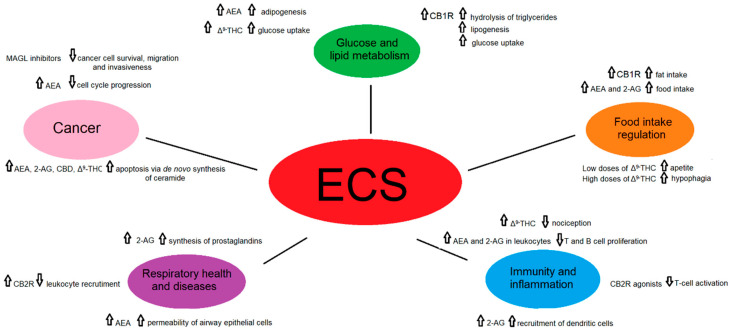
A schematic representation of the role of the endocannabinoid signaling in selected physiological and pathological states.

**Table 1 ijms-22-03661-t001:** Involvement of the ECS in various physiological and pathological processes.

Process	ECS Component	Metabolic Pathway and/or Effect	Reference
Glucose and lipid metabolism	CB1R stimulation in adipose tissue	increases the activity of the LPL, promotes hydrolysis of triglycerides, favoring lipogenesis through activation of lipogenic enzymes, and inhibition of the activity of the 5′-AMPK and promoting glucose uptake by translocation of the GLUT4	[22,23,24,25]
	AEA in adipose tissue	acts as a PPARγ agonist, amplifying the adipogenesis caused by the ECS	[26,27]
	CBD blocks CB1R	produces anti-obesity effects and relieves the symptoms of insulin resistance, type 2 diabetes and metabolic syndrome	[28]
	Δ^9^-THC	decreases TAGs and improves glucose uptake by an enhanced GLUT4 and IRS-2 gene expressions	[28,29]
	Endocannabinoids in the pancreas	important role in the regulation of cell proliferation and classification of α/β cell during pancreatic islets formation	[30,31]
	Activation of hepatic CB1R by endocannabinoids	induces the expression of ACC1, FAS and SREBPF1, resulting in fatty acid synthesis and leading to hepatic steatosis	[32]
	JD5037 and AM6545 in genetically and diet-induced obese mice	reduction of obesity, reverse leptin resistance and improve dyslipidemia, hepatic steatosis and insulin resistance and preserve beta cell function	[30,33,34,35]
Food intake regulation	Gastric CB1R activation by fat intake	increases fat-taste perception and promotes fat intake by ghrelin secretion	[30,36]
	AEA and 2-AG in plasma in humans	acts to initiate the intake and maintains the intake, respectively	[30,37,38,39]
	JD5037 in obese mice	diminishes leptinemia and reverses leptin resistance, resulting in a decrease in body weight and food intake	[33]
	AM6545 in obese mice	reduces obesity by reversing leptin resistance and improving dyslipidemia, insulin resistance and hepatic steatosis	[30,33,34,35]
	Low doses of Δ^9^-THC	suppression of glutamatergic transmission of CB1R and a rise of appetite	[30,40]
	High doses of Δ^9^-THC	GABAergic transmission of CB1R is disturbed, resulting in hypophagia	[30,40]
	Diminished anandamide levels by food enriched in n-3 PUFA	improves the lipid profile in obese subjects, preventing and treating metabolic disorders	[41,42]
Immunity and inflammation	CP55940 acting through CB2R in bone marrow	retention of immature B cells producing a significant decrease in CXCR4	[43,44,45]
	2-AG via activation of CB2R	recruits dendritic cells and their precursors during the innate immune response	[43,46]
	Cannabinoids acting through CB2R	inhibition of T-cell activation	[47,48,49]
	AEA and 2-AG in leukocytes	anti-inflammatory effects by decreasing T and B cell proliferation and proinflammatory and anti-inflammatory functions, respectively	[50,51]
	Rimonabant acting through CB1R	Rimonabant acting through CB1R	[52,53,54,55]
	CB2R activation	enhances fat tissue inflammation	[52,56,57]
	Increased levels of AEA and 2-AG in the brain with FAAH and MAGL	effective control of the immune response in different models of MS, HD and AD	[52,58]
	Δ^9^-THC in acute, visceral, inflammatory, and chronic pain	reduction of nociception	[59,60]
	CBD acting through CBR	inhibition of AEA enzymatic hydrolysis, activation/desensitization of TRPV1 and TRPA1 channels and inhibition ENTs, causing analgesia, and inhibition of inflammation	[60,61]
Respiratory health and diseases	AEA through non-canonical bioactive arachidonic metabolite formation	increases the permeability of airway epithelial cells	[62]
	2-AG	source of lung prostaglandins which metabolize into leukotriene B_4_ and C_4_ by neutrophils and eosinophils	[63,64]
	CB2R activation	inhibition of leukocyte recruitment and secretion of pro-inflammatory cytokines as TNF-α, IL-1β, IL-6, reduction of the formation of reactive oxygen species	[65]
	Inhibition of MAGL and FAAH	downregulation of TNF-α, PGE2, COX-2, iNOS	[66,67]
Cancer	AEA through CB1R and CB2R-CXCR4 heteromers in breast cancer	cell cycle progression blocking, inhibition of chemotaxis	[68,69,70]
	AEA, 2-AG, CBD, Δ^9^-THC in glioma	induction of apoptosis via de novo synthesis of ceramide, inhibition of cell migration and invasiveness through CB1R and CB2R activation	[71,72,73,74,75]
	AEA, 2-AG, Δ^9^-THC, CBD, HU-331, CP 55,940 acting through CB1R, CB2R and PPARγ in gastrointestinal tumors	cell invasiveness through the AKT/GSK3β signaling axis, induction of apoptosis through the inhibition of RAS–MAPK, PI3K–AKT and increased ceramide synthesis	[76,77,78,79]
	phyto-, endo- and synthetic cannabinoids, and MAGL inhibitors acting through CB1R, CB2R and CB2R-CXCR4 in prostate cancer	inhibition of cancer cell survival, migration and invasiveness through adenylyl cyclase, protein kinase A, EGFR	[69,80,81,82]
	CBD and CBR over-expression in lung cancer	activation of apoptosis, inhibition of ERK, PI3K, p38 MAPK, Akt, EGFR and ceramide-related pathways, tumor suppression by regulation of angiogenesis through up-regulation of PPAR-γ and cyclooxygenase-2	[83,84,85]

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
