# Peer review of "Functional Fine-Tuning of Metabolic Pathways by the Endocannabinoid System—Implications for Health and Disease"

_ijms, 2021, doi:10.3390/ijms22073661_

Round 1

Reviewer 1 Report

This is a comprehensive review that focus on the involvement of the endocannabinoid system (ESC) in functional fine-tuning of metabolic pathways. This work aims to give an overview about endocannabinoid system and metabolism in patho-physiological conditions to increase the growing burden of evidence about pharmacological use of ECS.

The authors have well tied and presented different aspects of ESC in various physiological and pathological processes such as glucose and lipid metabolism, food intake regulation, immunity and inflammation, cancer.

Some of the comments for the paper are as below:

  1. The introduction section could give a more detailed information about cannabinoids classification and related pathways (canonical and non-canonical)
  2. Authors could replace histograms in  figure 1 with a schematic pictorial model of CB1R and CB2R expression in human cells and tissues. It would be helpful for a better readership of paragraph n.2
  3. It would be good to include a small section on cannabinoids influence in differentiation processes focusing on stem-cell like and cancer stem cells, to carefully evaluate the involvement (role) of ESC in mechanism of cancer progression and resistance.
  4. The manuscript contains many sentences difficult to follow. Minor sentence formation and word repetition check required, for example check the sentences starting from the line 52 and 62.
  5. There are multiple alignment errors in Table 1, e.g. check reference number in page 5, cell alignment in “Metabolic pathway and/or effect” column.
  6. There are several typing/editing errors, check carefully the manuscript:

    lane 134: “2-Arachidonoylglycerol (2-AG),” it is underlined and written in a smaller font, 

    lane 136: fatty acid amide hydrolase

    lane 255: meal

    lane 256: nevertheless

    Abbreviations section style should be standardized 

Author Response

Answer: We thank the reviewer for the positive and constructive comments.

Some of the comments for the paper are as below:

  1. The introduction section could give a more detailed information about cannabinoids classification and related pathways (canonical and non-canonical)

Answer: More information about cannabinoids classification and related pathways (canonical and non-canonical) has been added to the Introduction section (page 2 lines 46-52 of the revised Manuscript)

  1. Authors could replace histograms in figure 1 with a schematic pictorial model of CB1R and CB2R expression in human cells and tissues. It would be helpful for a better readership of paragraph n.2

Answer: We thank the reviewer for this useful suggestion. We have added an additional figure as a schematic pictorial model of CB1R and CB2R expression in human organs and tissues for easier reading (Figure 1, page 2/3, lines 82-93 of the revised Manuscript)

  1. It would be good to include a small section on cannabinoids influence in differentiation processes focusing on stem-cell like and cancer stem cells, to carefully evaluate the involvement (role) of ESC in mechanism of cancer progression and resistance.

Answer: A section on cannabinoids influence in differentiation processes focusing on stem-cell like and cancer stem cells has been added to section 3.5 (pages 17/18 lines 508-514 of the revised Manuscript)

  1. The manuscript contains many sentences difficult to follow. Minor sentence formation and word repetition check required, for example check the sentences starting from the line 52 and 62.

Answer: We have reformatted/simplified the sentences and checked the manuscript for word repetition (page 2 lines 61-71, page 5 line 133-134 of the revised Manuscript)

  1. There are multiple alignment errors in Table 1, e.g. check reference number in page 5, cell alignment in “Metabolic pathway and/or effect” column.

Answer: Table 1 has been reformatted so there are no more alignment errors (pages 6-8 line 154 Table 1 of the revised Manuscript)

  1. There are several typing/editing errors, check carefully the manuscript:

lane 134: “2-Arachidonoylglycerol (2-AG),” it is underlined and written in a smaller font,

lane 136: fatty acid amide hydrolase

lane 255: meal

lane 256: nevertheless

Answer: Typing/editing errors have been corrected (page 12 line 203, page 13 lines 258/271/272/273/276/303, page 14 line 337, page 15 lines 377/378/390/392/395, page 16 line 457 of the revised Manuscript). “2-Arachidonoylglycerol (2-AG),” which was underlined and written in a smaller font (line 134) and fatty acid amide hydrolase (line 136) were a part of the Table abbreviations section that has been deleted following suggestions of other reviewers.

  1. Abbreviations section style should be standardized

Answer: The Abbreviations section style has been standardized and reformatted (page 18 line 521 of the revised Manuscript)

Reviewer 2 Report

This article deals with the homeostatic regulation exerted by the endocannabinoid system on several body functions. The emphasis was paid in this role in physiological conditions, but in particular in several pathological states related to energy metabolism, immunity and inflammation, cancer and others. Although numerous reviews on similar topics have been published in the last years, the idea is interesting and has been presented under a novel perspective that emphasizes the function of this regulatory system in the health and the disease. I have detected only a few aspects, most of them related to format and grammar questions, which would be susceptible to be improved. They are listed below:

  1. The abbreviations need to be revised. Given that a list is included at the end of the text, I think that there is no need to describe the abbreviation the first time that is used. Anyway, several abbreviations are described twice or even more times in the text, or they are described after having been used in abbreviated form before. This should be checked.
  2. Extending the above comment, THC should be written always (in the text and the tables) as Δ9-THC in order to differentiate this phytocannabinoid from its isomer Δ8-THC.
  3. In page 2 (but also in other parts of the text), N-acylethanolamines (and equivalent molecules) should be written with “N” in italics always. Also in this page 2 (line 57) and so on (page 12, line 380), prostaglandin glyceryl esters. In line 70, N-arachidonoyl glycine.
  4. Page 6, Table 1: Rimonabant, not Rimonavant.
  5. Page 9, lines 216-236: other line of research for this question is the design and development of CB1 receptor antagonists devoid of inverse agonist properties. This should be indicated.
  6. Page 11, line 363: fatty acid amide hydrolase should not be written in italics.
  7. I would recommend to add some diagrams or schemes describing in graphical form the role of the endocannabinoid signaling in some of the pathologies reviewed.

Author Response

Answer: We thank the reviewer for the positive and constructive comments.

I have detected only a few aspects, most of them related to format and grammar questions, which would be susceptible to be improved. They are listed below:

  1. The abbreviations need to be revised. Given that a list is included at the end of the text, I think that there is no need to describe the abbreviation the first time that is used. Anyway, several abbreviations are described twice or even more times in the text, or they are described after having been used in abbreviated form before. This should be checked.

Answer: The Abbreviations section style has been standardized (page 18 lines 535 of the revised Manuscript) and several abbreviations that have been described more times in the text or were described after having been used in abbreviated form before have been deleted (page 2 line 62/70, page 6 line 154, page 13 line 298, page 17 line 497, and Table 1 of the revised Manuscript). We believe that the abbreviations need to be defined in the text upon first mention for easier reading, but we have deleted the abbreviation list beneath Table 1 (page 11 lines 157-164 of the revised Manuscript).

  1. Extending the above comment, THC should be written always (in the text and the tables) as Δ9-THC in order to differentiate this phytocannabinoid from its isomer Δ8-THC.

Answer: THC has been re-written consistently (in the text and Table 1) as Δ9-THC (page 12 line 214, page 13 line 298/301, page 17 line 469/476/478/488 of the revised Manuscript)

  1. In page 2 (but also in other parts of the text), N-acylethanolamines (and equivalent molecules) should be written with “N” in italics always. Also in this page 2 (line 57) and so on (page 12, line 380), prostaglandin glyceryl esters. In line 70, N-arachidonoyl glycine.

Answer: The molecules in question have been written as suggested (page 2 line 63/77, page 15 line 393, Abbreviations section of the revised Manuscript)

  1. Page 6, Table 1: Rimonabant, not Rimonavant.

Answer: Rimonavant, has been replaced with Rimonabant (page 7 Table 1 of the revised Manuscript)

  1. Page 9, lines 216-236: other line of research for this question is the design and development of CB1 receptor antagonists devoid of inverse agonist properties. This should be indicated.

Answer: More information about CB1 receptor ligands has been added to the Section 3.1. (page 12, lines 245-247 of the revised Manuscript)

  1. Page 11, line 363: fatty acid amide hydrolase should not be written in italics.

Answer: This formatting error has been corrected (page 15 line 377 of the revised Manuscript)

  1. I would recommend to add some diagrams or schemes describing in graphical form the role of the endocannabinoid signaling in some of the pathologies reviewed.

Answer: Figure 3 has been added to describe in graphical form the role of the endocannabinoid signaling described in this manuscript (page 5 lines 134-137 of the revised Manuscript)

Reviewer 3 Report

This manuscript is a review article focused on role of ECS in glucose and lipid metabolism, food intake, immune homeostasis and inflammation, respiratory health and cancer. The review reports many evidences on the role played by ECS in the regulation of metabolism and in the maintenance of the homeostasis.  The review is timely and covers more of 190 articles. The manuscript is interesting but some paragraphs should be reorganized and shortened, considering the information already reported previously.

In details:

In the introduction some references should be reported, for example for the pathways activated by ECs.

It is difficult to understand if the data reported are referred to humans or mice or other mammals. Please indicate this.

Paragraph 3.2: lines 248-252 “In addition to promoting energy intake, ECS participates in the control of lipid and glucose metabolism in several peripheral organs, in particular in adipose tissue and in the liver, and also direct actions in pancreas and in skeletal muscle, in order to maintain metabolic homeostasis. This knowledge may help in the design of future therapies for the metabolic syndrome”.

This sentence could be deleted or better explained since it seems do not completely fit in the context “food intake regulation”.

Paragraph 3.3 Immunity and inflammation is confused. It should be reorganized and  rewritten. The sentence : “ECS is involved in glucose and lipid metabolism [56,98,101]. Thus far, abnormal endocannabinoid signalling, either due to their excessive production or to up-regulation of CB1R, which exerts damaging effects in diabetes, is considered a pathogenic factor in this disease. Recent studies shows that the pharmacological blockade of CB1R prevents metabolic dysfunction and ß-cell loss, while reducing body mass and mortality rate [56,143]. Please explain the meaning of the sentence in the context of this paragraph.

Author Response

Answer: We thank the reviewer for the positive and constructive comments.

In details:

  1. In the introduction some references should be reported, for example for the pathways activated by ECs.

Answer: References for the pathways activated by the ECS have been reported in the introduction section (page 2 line 45/50 of the revised Manuscript)

  1. It is difficult to understand if the data reported are referred to humans or mice or other mammals. Please indicate this.

Answer: Further clarification that the data refers to humans has been introduced (page 2 line 74/80, page 5 line 110 and line 126 of the revised Manuscript)

  1. Paragraph 3.2: lines 248-252 “In addition to promoting energy intake, ECS participates in the control of lipid and glucose metabolism in several peripheral organs, in particular in adipose tissue and in the liver, and also direct actions in pancreas and in skeletal muscle, in order to maintain metabolic homeostasis. This knowledge may help in the design of future therapies for the metabolic syndrome”.

This sentence could be deleted or better explained since it seems do not completely fit in the context “food intake regulation”.

Answer: We thank the reviewer for the constructive comment, this sentence was deleted (page 13 lines 266-269 of the revised Manuscript)

  1. Paragraph 3.3 Immunity and inflammation is confused. It should be reorganized and rewritten. The sentence: “ECS is involved in glucose and lipid metabolism [56,98,101]. Thus far, abnormal endocannabinoid signalling, either due to their excessive production or to up-regulation of CB1R, which exerts damaging effects in diabetes, is considered a pathogenic factor in this disease. Recent studies shows that the pharmacological blockade of CB1R prevents metabolic dysfunction and ß-cell loss, while reducing body mass and mortality rate [56,143]. Please explain the meaning of the sentence in the context of this paragraph.

Answer: We thank the reviewer for the constructive comments, this sentence was rewritten to improve the 3.3 Immunity and inflammation section (pages 14 and 15 lines 354-358 and 362/366 of the revised Manuscript). Diabetes is an inflammatory disease and targeting inflammatory pathways could possibly be a component of the strategies to prevent and control diabetes and related complications.